# Assessment of Radioactive Materials in Albite Granites from Abu Rusheid and Um Naggat, Central Eastern Desert, Egypt

**Ibrahim Gaafar** [1,†], **Mona Elbarbary** [1], **M. I. Sayyed** [2,3], **Abdelmoneim Sulieman** [4], **Nissren Tamam** [5], **Mayeen Uddin Khandaker** [6], **David A. Bradley** [6,7] **and Mohamed. Y. Hanfi** [1,8,*,†]

1    Nuclear Materials Authority, P.O. Box 530, El Maadi, Cairo 11936, Egypt; gaafar_ibrahim@yahoo.com (I.G.); monaelbarbary@gmail.com (M.E.)
2    Department of Nuclear Medicine Research, Institute for Research and Medical Consultations, Imam Abdulrahman Bin Faisal University, Dammam 31441, Saudi Arabia; mabualssayed@ut.edu.sa
3    Department of Physics, Faculty of Science, Isra University, Amman 11622, Jordan
4    Department of Radiology and Medical Imaging, College of Applied Medical Sciences, Prince Sattam bin Abdulaziz University, P.O.Box 422, Alkharj 11942, Saudi Arabia; a.sulieman@psau.edu.sa
5    Department of Physics, College of Science, Princess Nourah bint Abdulrahman University, P.O. Box 84428, Riyadh 11671, Saudi Arabia; nmtamam@pnu.edu.sa
6    Centre for Applied Physics and Radiation Technologies, School of Engineering and Technology, Sunway University, Bandar Sunway 47500, Selangor, Malaysia; mayeenk@sunway.edu.my (M.U.K.); d.a.bradley@surrey.ac.uk (D.A.B.)
7    Centre for Nuclear and Radiation Physics, Department of Physics, University of Surrey, Guildford GU2 7XH, UK
8    Institute of Physics and Technology, Ural Federal University, Mira Street 19, 620002 Ekaterinburg, Russia
*    Correspondence: mokhamed.khanfi@urfu.ru; Tel.: +7-98-2646-6357
†    These authors contributed equally to this work.

**Abstract:** The present study aims to assess Abu Rusheid and Um Naggat albite granite's natural radioactivity in the Central Eastern Desert, Egypt, using an HPGe laboratory spectrometer. A total of 17 albite granite samples were detected for this study. The activity concentrations were estimated for $^{238}$U (range from 204 to 1127 Bq/kg), $^{226}$Ra (range from 215 to 1300 Bq/kg), $^{232}$Th (from 130 to 1424 Bq/kg) and $^{40}$K (from 1108 to 2167 Bq/kg) for Abu Rusheid area. Furthermore $^{238}$U (range from 80 to 800 Bq/kg), $^{226}$Ra (range from 118 to 1017 Bq/kg), $^{232}$Th (from 58 to 674 Bq/kg) and $^{40}$K (from 567 to 2329 Bq/kg) for the Um Naggat area. The absorbed dose rates in the outdoor air were measured with average values of 740 nGy/h for Abu Rusheid albite granite and 429 nGy/h for Um Naggat albite granite. The activity concentration and gamma-ray exposure dose rates of the radioactive elements $^{238}$U, $^{226}$Ra, $^{232}$Th and $^{40}$K at Abu Rusheid and Um Naggat exceeded the worldwide average values that recommend the necessity of radiation protection regulation. Moreover, the corresponding outdoor annual effective dose (AED$_{out}$) was calculated to be 0.9 and 0.5 mSv y$^{-1}$ for Abu Rusheid and Um Naggat albite granite, respectively, which are lower than the permissible level (1 mSv y$^{-1}$). By contrast, the indoor annual effective dose (AED$_{in}$) exceeded the recommended limit (3.6 and 2.1 for Abu Rusheid and Um Naggat, respectively). Therefore, the two areas are slightly saving for development projects concerning the use of the studied rocks. The statistical analysis displays that the effects of the radiological hazard are associated with the uranium and thorium activity concentrations in Abu Rusheid and Um Naggat albite granites.

**Keywords:** albite granites; gamma-ray spectrometer; mineral analysis; radiological hazard indices; natural radioactivity

## 1. Introduction

Natural radioactivity in the environment is due to the presence of natural radionuclides, namely $^{238}$U, $^{232}$Th and $^{40}$K, in various geological formations [1,2]. Terrestrial radiation comprises radiation emitted from these radionuclides and their progeny. $^{40}$K

is a singly occurring natural radionuclide, which also emits gamma radiation. In total, 98.5% of the gamma dose received from $^{238}$U series is emitted from $^{226}$Ra and its daughter products [3–5]. Because the ornamental and construction materials used in buildings are derived from natural rocks, such as granitic rocks, it is critical to assess the radiological dangers that humans face by measuring the concentration levels of radionuclides in these rocks. Furthermore, radionuclide concentration data will aid in the creation of guidelines and standards for the use and management of these rocks [6]. Geological materials utilized in industry, such as granitic rocks and their industrial derivatives, as well as concrete, cement, brick, sand, aggregate, marble, granite, limestone gypsum, etc. typically include a small quantity of terrestrial radioisotopes in varying quantities, depending on the origin of the rocks. As a result, a thorough study of the concentrations and distributions of terrestrial isotopes in rocks is required to avoid using geological materials with high levels of terrestrial radioisotopes, which could result in natural radioactivity contaminating the environment [7–10].

According to the ATSDR (Agency for Toxic Substances and Disease Registry), long-term radioactive exposure causes significant ailments that include chronic lung disease, oral necrosis, leukopenia, and anemia [11,12]. In recent years, multiple investigations on high natural background areas around the world have created more awareness of risk appraisal due to whole-body exposures to long-term low-level radiation among inhabitants [13]. High radiation levels are caused by the presence of radionuclides in high concentrations in granite rocks, soils, sediments, and other materials. Granites are igneous rocks that were formed by slowly cooling magma. Distinct types of granite feature different geologic origins and mineralogic compositions. Granitic rocks contain different amounts of naturally occurring radionuclides of terrestrial origin, $^{238}$U and $^{232}$Th series and $^{40}$K [14,15]. Granites from Abu Rusheid and Um Naggat include specific mineralogical and geochemical features. These granites are acidic, slightly per-aluminous-to-meta-aluminous, Li–F–Na-rich, and Sn–Nb–Ta-mineralized. Snowball textures, homogenous distribution of rock-forming accessory minerals, disseminated mineralization and melt inclusions in quartz phenocrystals are typical features indicative of their petrographic specialization [16,17]. The late magmatic to the early hydrothermal stage is represented by small crystallized silicic melt in greisen and the outer margins of the mineralized veins. These inclusions are associated with beryl, topaz, and cassiterite mineralization.

The present study aimed to detect the radionuclides levels in the examined albite granites in particular, since they may be used in building materials and infrastructural applications. The studied areas were selected for the investigation owing to the economic importance of the heavy minerals accumulated in the albite granites. Furthermore, the public exposure to radiation was evaluated by assessing radiological hazards with various radioactive parameters. Among the radiological hazards, the parameters included: radium equivalent activity ($Ra_{eq}$); the absorbed dose rate ($D_R$); the annual effective dose (AED), both outdoor and indoor; and external ($H_{ex}$) and internal ($H_{in}$) hazard indices. Moreover, the excess lifetime cancer risk (ELCR) and the annual gonadal dose equivalent (AGDE) were also estimated.

## 2. Materials and Methods

### 2.1. Geologic Setting

#### 2.1.1. Um Naggat Area

Um Naggat albite granites (Figure 1) are concentrically zoned, elliptically shaped granitic bodies composed of a central biotite monzogranite and an outer alkaline-to-peralkaline granite rim with hypersolvus textures, with biotite, allanite, cassiterite and fluorite. The northern rim of the granite is albitized. Alterations are developed along with a NW-trending structure, and kaolinzation, greisenization alterations and yellow-to-violet fluorite veins are recorded [18]. Pegmatite lenses are characterized by garnet, fluorite and Nb-Ta mineralization occurring at the northern margin. The high Nb (104 to 672 ppm), Zr

(223 to 2022 ppm), and Y (35 to 180 ppm) contents of the two granites indicate that they both possess alkaline-to-peralkaline features [18,19].

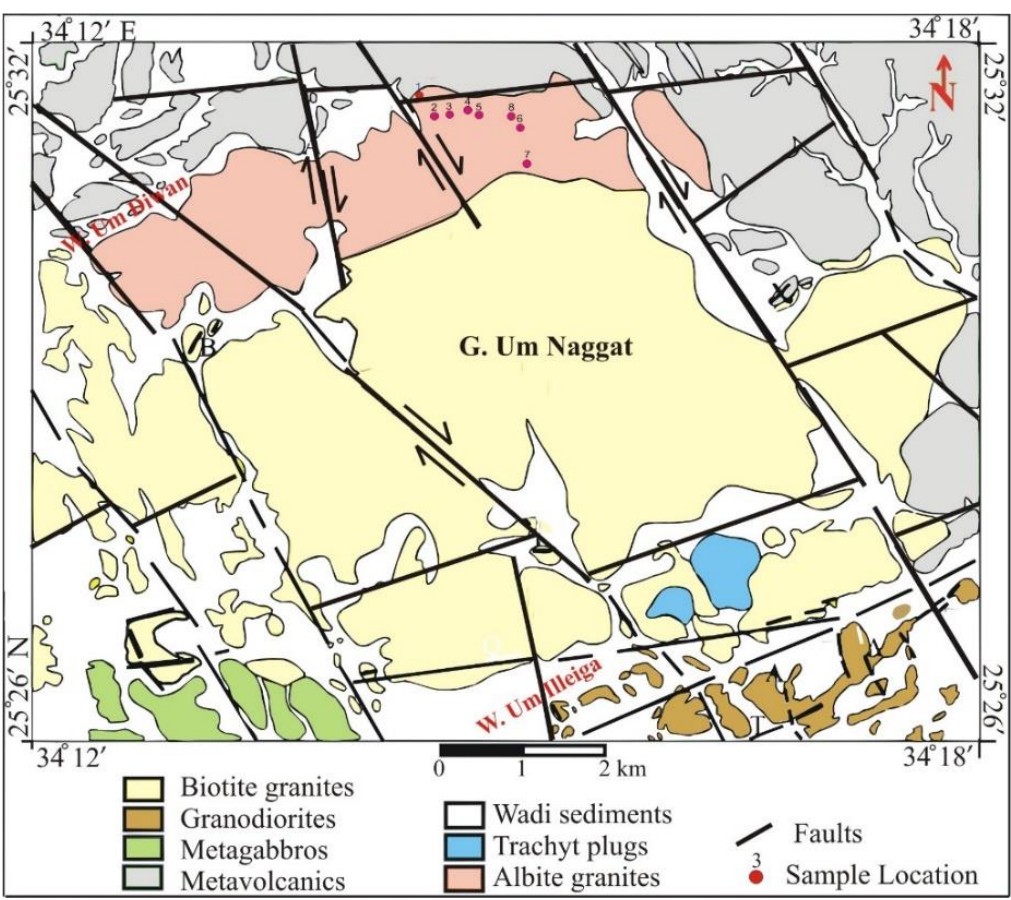

**Figure 1.** Geologic map of Um Naggat area, Central Eastern Desert, Egypt.

### 2.1.2. Abu Rusheid Area

The Abu Rusheid area is a part of the Arabian-Nubian Shield that lies in the Wadi Al Gemal basin and can be considered a key domain in that shield, besides its very complex structures. The main rock units encountered in the Abu Rusheid area are represented by albite granites, quartz amphibole schist, amphibolites, and gneisses (Figure 2). This area is located between two major thrusts in the NE. The Abu Rusheid area features abundant zircon, fluorite, and columbite [20]. Columbite features an intermediate composition between Fe- and Mn-columbite end-members and variable U contents (0.03 to 1.41 wt% $UO_2$) [21]. Thorite has also been observed. Sulphides may also be locally abundant (pyrite, chalcopyrite, and arsenopyrite). This granite's Nb/Ta ratio is close to the average crustal ratio. Previously interpreted as metasomatized psammitic gneisses, these rocks feature all the characteristics of rare-metal granite [21] as highly fractionated to alkaline magma.

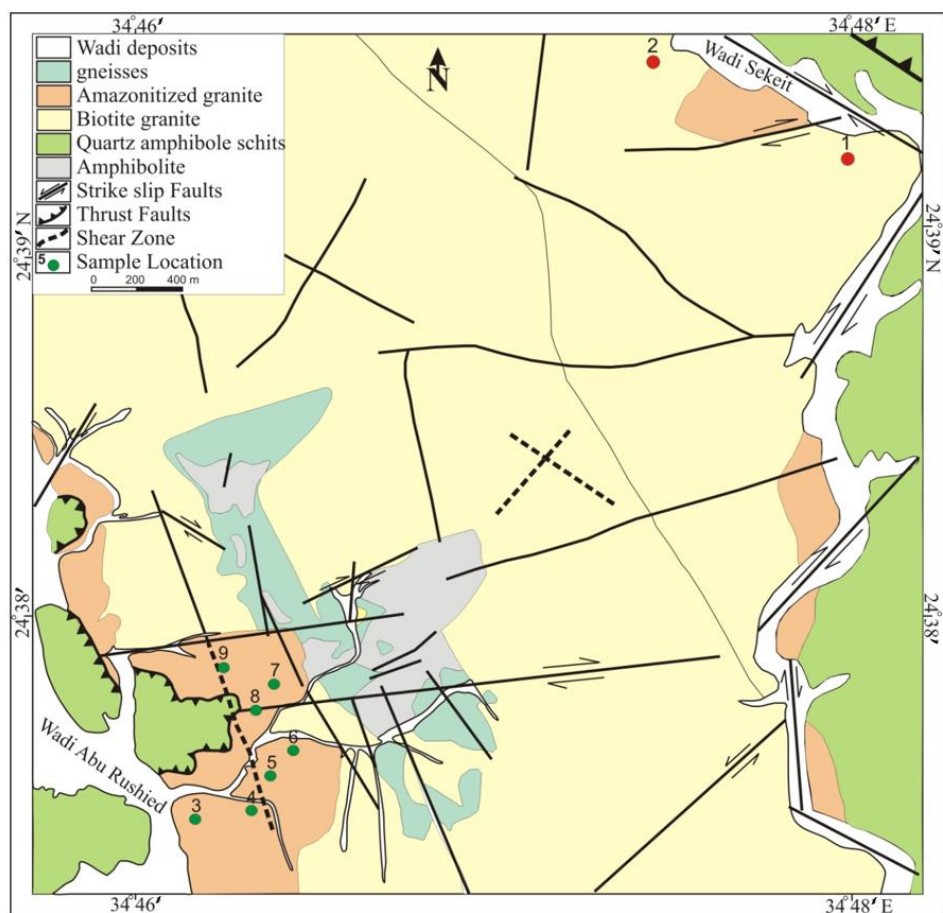

**Figure 2.** Geologic map of Abu Rusheid area, Central Eastern Desert, Egypt.

*2.2. Analytical Techniques*

The collected samples were transferred to the laboratory to evaluate their radiological hazards. Next, they were stored within the plastic containers for 28 days to achieve secular equilibrium between radon and its daughters. The activity of a radioactive source is defined as the rate at which the isotope decays. Radioactivity may be thought of as the quantity of radiation produced in a given amount of time. The radioactivity concentration of the different identified radionuclides was calculated by gamma-ray spectrometry HPGe with the following simple regression [22]. The calibration energy of the gamma spectrometer system HPGe used in the investigation was accomplished with a mixed source with the same shape as the samples. At the same time, an empty container was used to identify the background counting. The radionuclide activity concentrations were computed utilizing the following equation:

$$A = \text{Net area (cps)}/I_\gamma . \ \zeta. \ M \ \text{Bq/kg} \qquad (1)$$

where

A = The activity concentration of the gamma-ray spectral line in Bq/kg
Net area (cps) = The net detected counts corresponding to the energy per second.
$\zeta$ = The counting system efficiency of the energy.
M = The mass of the sample in kg.
$I_\gamma$ = The intensity of the gamma-ray spectrum.

In this study, 17 albite granite samples were collected from Abu Rusheid and Um Naggat. Natural radionuclides relevant to this work were mainly gamma-ray emitting nuclei in the decay series of $^{238}$U, $^{235}$U and $^{232}$Th, and singly occurring $^{40}$K. While $^{40}$K can be measured directly by its own gamma-rays, $^{238}$U, $^{235}$U, and $^{232}$Th are not directly

gamma-ray emitters, but it is possible to measure the gamma-rays of their decay products. Decay products for $^{238}$U ($^{234}$Th: 63.3; $^{234}$Pa: 1001; $^{214}$Pb: 295 and 352 keV; and $^{214}$Bi: 609, 1120, 1238, 1377 and 1764 Kev), $^{235}$U ($^{235}$U: 143,163,185 and 205 Kev) and $^{232}$Th ($^{228}$Ac: 209, 338, 911 and 968 keV; $^{212}$Bi: 727 keV; and $^{208}$Tl: 583 keV) were used by assuming the decay series to be in secular equilibrium [23]. $^{40}$K was determined as (1460 keV) photopeak. For the actinium series, $^{235}$U $\gamma$- energies of (143.8 keV, 163.4 keV) were taken to represent the $^{235}$U activity [24]. Weighted averages of several decay products were used.

## 3. Results and Discussion

### 3.1. Radiometric Evaluation of the Studied Areas

Gamma-ray spectrometry directly measures the surface distribution of naturally occurring radioelement K, U, and Th. Potassium is a major constituent of most rocks, while uranium and thorium are present in trace amounts as mobile and immobile elements, respectively. As the concentrations of these radioelements vary between different rock types, the measured radioelement distribution can be reliably used to map and distinguish different lithologies [25].

### 3.1.1. Abu Rusheid Area

Albite granite features medium thorium, potassium and uranium contents, which are radioactive elements. A small part of the amphibole rock in the northeast region of the studied area contains high concentrations of uranium and thorium with low potassium content. Some small parts of amazonitized granite contain high concentrations of uranium, up to 200 ppm, while their potassium and thorium contents are medium. Most of the amazonitized granites are characterized by very high concentrations of thorium, up to 500 ppm, and normal potassium contents. It is noted that the uranium and thorium anomaly trends are often affected by the north-east and north-west tectonic trends in the studied area. From Table 1, it is clear that the albite granite features slightly higher eU contents (20.3 to 263 ppm, average 118.2 ppm). At the same time, the albite granite contains higher concentrations of both thorium (56–552 ppm, average 305 ppm) and potassium (3.6–7.6%, average 5.2%).

**Table 1.** Local content of the radioactive elements for albite granites at Abu Rusheid area, South Eastern Desert, Egypt.

| Station | Location | K, % | eU, ppm | eTh, ppm | eTh/K |
|---------|----------|------|---------|----------|-------|
| Rsh1 | 24°39.475′ N 34°47.602′ E | 6.5 | 21.1 | 57 | 8.8 |
| Rsh 2 | 24°39.817′ N 34°46.878′ E | 6.3 | 36.3 | 68.3 | 10.8 |
| Rsh 3 | 24°37.864′ N 34°46.027′ E | 4.7 | 219.7 | 542 | 114.5 |
| Rsh 4 | 24° 37.908′ N 34°46.198′ E | 4.9 | 77.3 | 256 | 52.6 |
| Rsh 5 | 24°37.977′ N 34°46.220′ E | 4.8 | 191.3 | 481 | 99.5 |
| Rsh 6 | 24°38.057′ N 34°46.290′ E | 4.8 | 120 | 277.7 | 57.4 |
| Rsh 7 | 24°38.095′ N 34°46.091′ E | 4.9 | 191.3 | 392.3 | 80.1 |
| Rsh 8 | 24°38.074′ N 34°46.065′ E | 5.3 | 75.3 | 346.7 | 65.8 |
| Rsh 9 | 24°38.167′ N 34°45.940′ E | 4.6 | 131.3 | 324 | 69.9 |
| Ave | | 5.2 | 118.2 | 305 | 62.6 |

The potassium–thorium cross plot is widely used to recognize clay mineral associations and to discriminate micas and feldspars [26]. As both thorium (by adsorption) and potassium (chemical composition) are associated with clay minerals, the ratio eTh/K expresses relative potassium enrichment as an indicator of clay-mineral species and might be diagnostic of other radioactive minerals [27,28]. Therefore, if eTh/K $\geq 2 \times 10^{-4}$, the rock is thorium-rich, and if eTh/K $\leq 1 \times 10^{-4}$, the rock is potassium-rich [29,30]. Table 1 presents the results for the Abu Rusheid area, in which the eTh/K ratio ranged from 8.4 to 48.3, with an average value of 62.6. These values are much higher than $2 \times 10^{-4}$, which indicates that this area's albite granite is fresh.

### 3.1.2. Um Naggat Area

Considering the surface distribution of radioactive elements (K, eU, eTh) in Table 2, it is clear that Um Naggat albite granite features eU contents (13 to 105 ppm, average 54 ppm) slightly, while the albite granite contains concentrations of both thorium (35 to 280 ppm, average 97 ppm) and potassium (1.7 to 8%, average 4.5%). The eTh/K ratio ranges from 4.5 to 137.8, with an average value of 31.3. These values are much higher than $2 \times 10^{-4}$, which indicates that this area's albite granite is fresh.

**Table 2.** Local content of the radioactive elements for albite granites at Um Naggat area, Central Eastern Desert, Egypt.

| Station | Location | K | eU | eTh | eTh/K |
|---------|----------|-----|------|------|-------|
|  |  | % | ppm | ppm |  |
| Ung 1 | 25°31.146′ N 34°12.974′ E | 2 | 77 | 246 | 125.1 |
| Ung 2 | 25°31.009′ N 34°13.054′ E | 4.6 | 86.3 | 121.3 | 26.4 |
| Ung 3 | 25°31.018′ N 34°13.096′ E | 4.7 | 101.3 | 126.7 | 27 |
| Ung 4 | 25°31.043′ N 34°13.155′ E | 2.7 | 45.3 | 73.3 | 26.8 |
| Ung 5 | 25°31.015′ N 34°13.185′ E | 4.1 | 69.3 | 80.3 | 19.4 |
| Ung 6 | 25°30.960′ N 34°13.309′ E | 5.1 | 19.7 | 44.3 | 8.7 |
| Ung 7 | 25°30.794′ N 34°13.330′ E | 5 | 16.3 | 37.3 | 7.5 |
| Ung 8 | 25°31.008′ N 34°13.286′ E | 7.5 | 16.3 | 47 | 6.3 |
| Ave. |  | 4.5 | 54 | 97 | 31.3 |

Figures 3 and 4 exhibit the geological map of radioactive contents (eU, eTh and K%) in the albite granite samples. As can be seen in Figure 3, the highest uranium and thorium contents observed in the studied stations are assembled in the south-west of the investigated area. By contrast, Figure 4 shows that uranium and thorium contents are predicted in the south and south-east of the studied stations in the Um Naggat area. This is due to the alteration the radioactive minerals deposited inside the cracks of the granites. Furthermore, the highest thorium activity concentrations were found in some parts of the studied area. This was due to the presence of different minerals, such as zircon, thorianite, and monazite in the granite samples. The ratio $^{238}$U/$^{232}$Th demonstrates that the granite enriched with the uranium due to the leaching process from rainwater helped in the migration of uranium minerals (uranophane, uraninite, betauranophane) and precipitated at joints and faults [31].

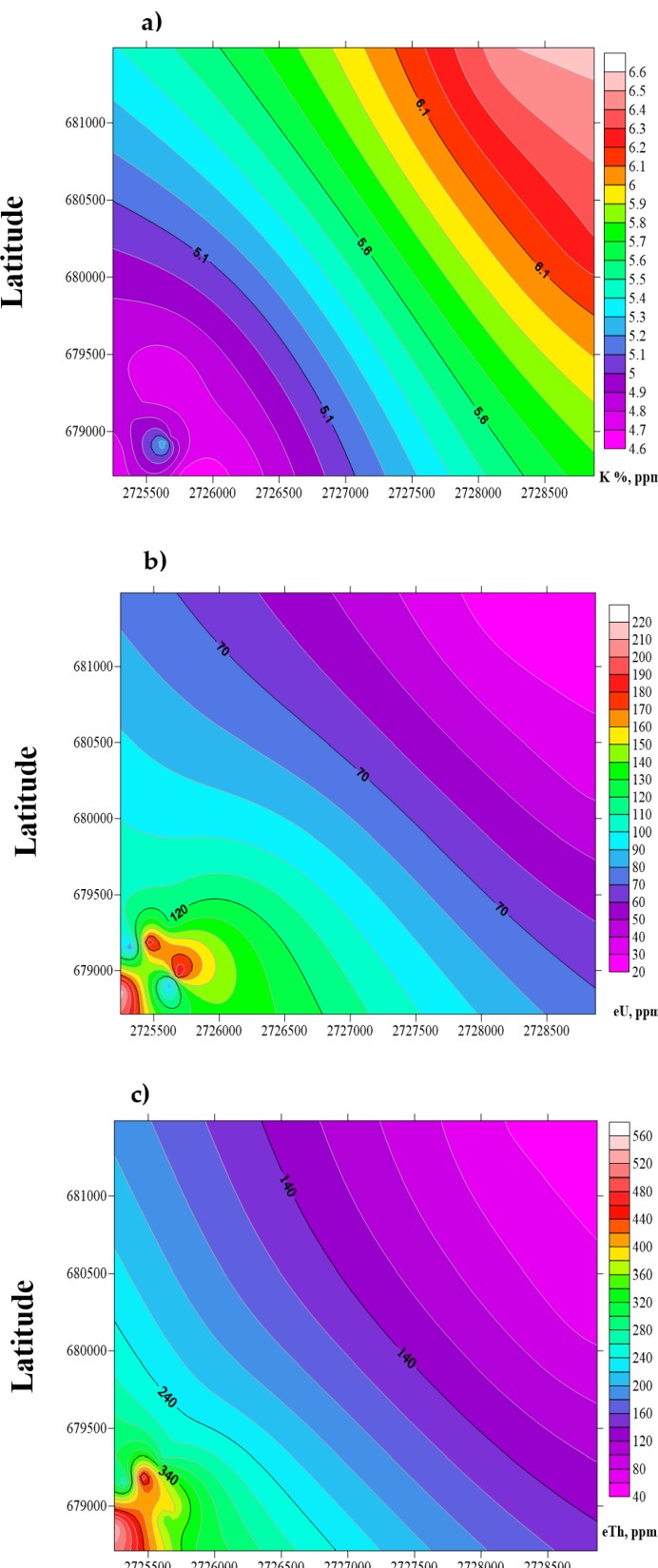

**Figure 3.** The geological map described the accumulation of high concentrations of: (**a**) potassium, K%, (**b**) uranium, eU, and (**c**) thorium, eTh of the albite granites in the Abu Rusheid area.

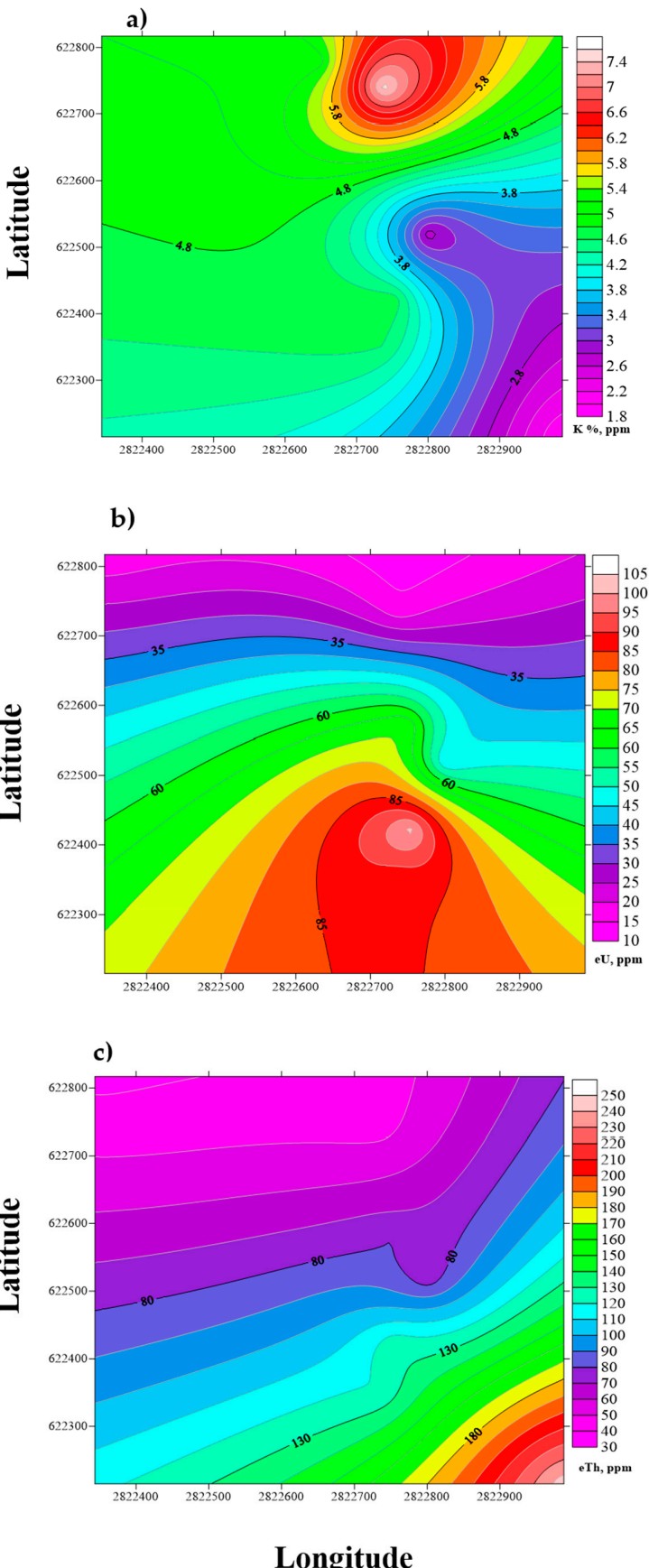

**Figure 4.** The geological map describing the accumulation of high concentrations of (**a**) potassium, K%, (**b**) uranium, eU, and (**c**) thorium, eTh of the albite granites in the Um Naggat area.

### 3.2. Radiometric Evaluation of the Studied Areas

Table 3 shows that the activity concentration of the $^{238}$U series included $^{234m}$Pa that varied from 217.9 to 1135 Bq/kg and $^{234}$Th that varied from 189.8 to 1119.7 Bq/kg. Meanwhile, in Table 4, the $^{232}$Th series presents values of $^{228}$Ac ranges from 124.8 to 1324.7 Bq/kg, $^{208}$Ti ranges from 132.1 to 1389.1 Bq/kg, and $^{212}$Bi ranges from 133.3 to 1559 Bq/kg.

**Table 3.** Activity concentration of $^{238}$U (Bq/kg) isotopes for Abu Rusheid samples, South Eastern Desert, Egypt.

| Sample | $^{234}$Th | $^{234}$Pa | $^{226}$Ra | $^{214}$Pb | $^{214}$Bi | $^{230}$Th | $^{210}$Pb |
|---|---|---|---|---|---|---|---|
| Rsh 1 | 189.8 | 217.9 | 214.8 | 186.5 | 169.3 | 121.7 | 142.5 |
| Rsh 2 | 241.2 | 230.3 | 262.7 | 220 | 200.7 | 365.4 | 210.4 |
| Rsh 3 | 476.7 | 476.2 | 521.7 | 476.8 | 428 | 449.5 | 329.6 |
| Rsh 4 | 307.3 | 330.1 | 341 | 301.1 | 291.8 | 475.5 | 210.4 |
| Rsh 5 | 1119.7 | 1135 | 1300.1 | 1173.3 | 1112 | 1410.5 | 894.6 |
| Rsh 6 | 728.4 | 661.4 | 872.9 | 763.7 | 722 | 825.8 | 558.5 |
| Rsh 7 | 472.8 | 431.5 | 555.2 | 465.7 | 424.7 | 458.8 | 396.7 |
| Rsh 8 | 472.7 | 381.8 | 497.2 | 398.9 | 372.7 | 559.4 | 261.7 |
| Rsh 9 | 644.3 | 515.7 | 634.5 | 554.4 | 503.8 | 700.5 | 363.9 |

**Table 4.** Activity concentration of $^{232}$Th (Bq/kg) isotopes of albite granites samples at Abu Rusheid, Central Eastern Desert, Egypt.

| Samples | $^{228}$Ac | $^{208}$Tl | $^{212}$Bi |
|---|---|---|---|
| Rsh 1 | 124.8 | 132.1 | 133.3 |
| Rsh 2 | 143 | 154.9 | 167.4 |
| Rsh 3 | 992.8 | 1061.7 | 1144.8 |
| Rsh 4 | 437 | 475.3 | 506.7 |
| Rsh 5 | 842 | 900.3 | 964.2 |
| Rsh 6 | 460 | 490.4 | 559.3 |
| Rsh 7 | 765.8 | 803.7 | 875.2 |
| Rsh 8 | 716 | 754.1 | 808.4 |
| Rsh 9 | 1324.7 | 1389.1 | 1559 |

As demonstrated in Table 5, the concentrations of all the samples were very high compared to the worldwide average, except $^{235}$U, which is lower than the worldwide average. The average worldwide radioelement values are 32 Bq/kg for $^{226}$Ra, 45 Bq/kg for $^{232}$Th, and 412 Bq/kg for $^{40}$K [2]. The $^{226}$Ra/$^{238}$U for the samples were around the unity, which suggested that Abu Rusheid granites are characterized by secular uranium equilibrium. The $^{238}$U/$^{235}$U activity ratio for all the samples varied between 21.1 and 21.8 in the Abu Rusheid albite granite, which agrees with the worldwide crustal average.

Figure 5 represents the correlation curves of the activity concentrations between $^{226}$Ra and $^{232}$Th, $^{226}$Ra and $^{40}$K, and $^{232}$Th and $^{40}$K for the Abu Rusheid area. Figure 5a shows a weak correlation between $^{226}$Ra and $^{232}$Th ($R^2$ = 0.21), which could be explained by the high U-enrichment. Furthermore, there is a weak correlation between $^{226}$Ra and $^{40}$K ($R^2$ = 0.35) (Figure 5b). Moreover, there is a negative correlation between $^{232}$Th and $^{40}$K ($R^2$ =0.51) due to the high enrichment of $^{232}$Th compared to the low $^{40}$K-content, as shown in Figure 5c. Moreover, the $^{232}$Th/$^{238}$U ratio for the Rsh (3,4,7,8,9) samples is higher than Clark's value (3.5), which indicates Th enrichment. By contrast, the samples Rsh 1, Rsh 2, Rsh 5, and Rsh 6 are less than Clark's value, indicating U-enrichment.

**Table 5.** Activity concentration of $^{238}$U, $^{226}$Ra, $^{235}$U, $^{232}$Th, and $^{40}$K in (Bq/kg) of albite granites for Abu Rusheid samples, South Eastern Desert, Egypt.

| Sample | $^{238}$U | $^{226}$Ra | $^{235}$U | $^{232}$Th | $^{40}$K | $^{238}$U/$^{235}$U | $^{226}$Ra/$^{238}$U | $^{232}$Th/$^{238}$U |
|---|---|---|---|---|---|---|---|---|
| Rsh 1 | 203.9 | 214.8 | 9.6 | 130.1 | 1751.5 | 21.2 | 1.1 | 0.64 |
| Rsh 2 | 235.7 | 262.7 | 11.1 | 155.1 | 2166.7 | 21.3 | 1.1 | 0.66 |
| Rsh 3 | 476.4 | 521.7 | 22.6 | 1066.4 | 1254.3 | 21.1 | 1.1 | 2.24 |
| Rsh 4 | 318.7 | 341 | 15 | 473 | 1277.4 | 21.3 | 1.1 | 1.48 |
| Rsh 5 | 1127.3 | 1300 | 53.2 | 902.2 | 1139.1 | 21.2 | 1.1 | 0.80 |
| Rsh 6 | 694.9 | 872.9 | 32.8 | 503.1 | 1152.9 | 21.2 | 1.3 | 0.72 |
| Rsh 7 | 452.1 | 555.2 | 20.8 | 814.9 | 1107.9 | 21.8 | 1.2 | 1.80 |
| Rsh 8 | 427.3 | 497.2 | 19.9 | 759.5 | 1176.2 | 21.5 | 1.1 | 1.78 |
| Rsh 9 | 580 | 634.5 | 26.6 | 1424.3 | 1147 | 21.8 | 1.1 | 2.46 |
| Min | 203.9 | 214.8 | 9.6 | 130.1 | 1107.9 | 21.1 | 1.1 | 0.64 |
| Max | 1127.3 | 1300 | 53.2 | 1424.3 | 2166.7 | 21.8 | 1.3 | 2.46 |
| Ave. | 501.8 | 577.8 | 23.5 | 692 | 1352.6 | 21.4 | 1.13 | 1.40 |
| W. A | 33 | 32 | 33 | 45 | 500 | 21.7 | 1 | 3.5 |

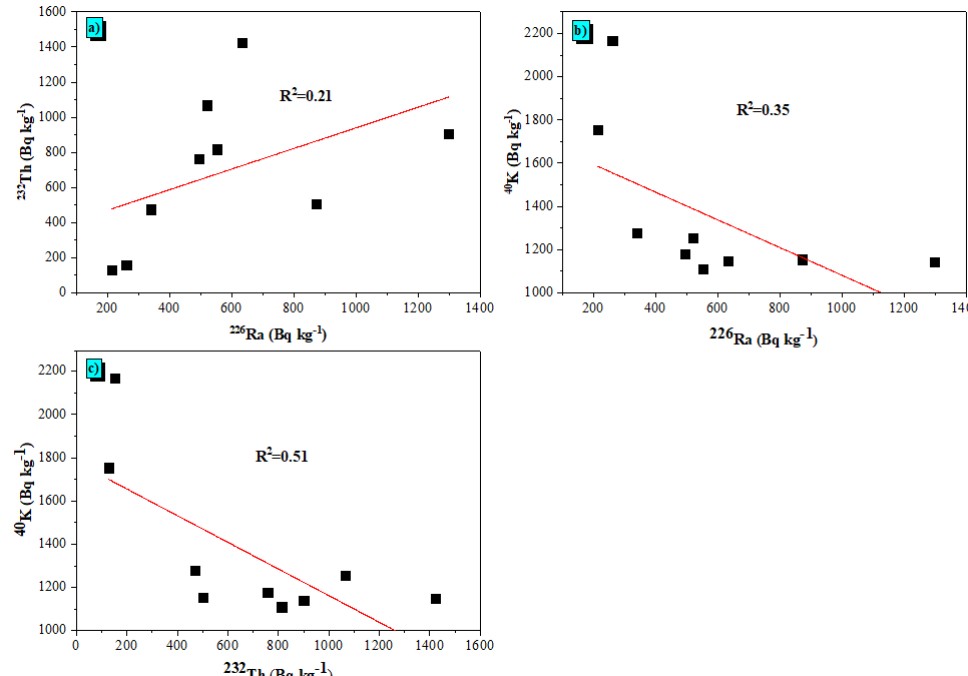

**Figure 5.** Correlation of the activity concentrations between: (**a**) $^{226}$Ra and $^{232}$Th (**b**) $^{226}$Ra and $^{40}$K, and (**c**) $^{232}$Th and $^{40}$K of albite granites in Abu Rusheid area, Central Eastern Desert, Egypt.

The concentrations and distribution of radionuclides in the eight studied samples from the um Naggat area were determined using an HPGe spectrometer to evaluate the environmental radioactivity. Table 6 shows the activity concentration of the $^{238}$U series of the Um Naggat albite granite samples. The activity concentrations of the $^{238}$U series included $^{234m}$Pa that varied from 79.3 to 753.7 Bq/kg, $^{234}$Th that varied from 80.9 to 847.2 Bq/kg. Meanwhile, the $^{232}$Th series features values of $^{228}$Ac ranges from 54.7 to 632.1 Bq/kg, $^{208}$Tl ranges from 56.9 to 668.8 Bq/kg and $^{212}$Bi ranges from 63.2 to 720.9 Bq/kg, as shown in Table 7.

**Table 6.** Activity concentration of $^{238}$U isotopes of albite granite samples from Um Naggat Central Eastern Desert, Egypt.

| Samples | $^{234}$Th | $^{234}$Pa | $^{226}$Ra | $^{214}$Pb | $^{214}$Bi | $^{230}$Th | $^{210}$Pb |
|---|---|---|---|---|---|---|---|
| UNg 1 | 834.7 | 735.5 | 844.5 | 738.7 | 726.8 | 641.7 | 575.8 |
| UNg 2 | 575.7 | 580.5 | 725.7 | 512.2 | 510.9 | 762.9 | 434.7 |
| UNg 3 | 442.1 | 415.1 | 514.9 | 437.9 | 428.1 | 633.4 | 340 |
| UNg 4 | 369.8 | 350.2 | 345.3 | 338.7 | 328.1 | 301.3 | 257.5 |
| UNg 5 | 847.2 | 753.7 | 1016.6 | 730.6 | 708.6 | 963.9 | 600.3 |
| UNg 6 | 125.5 | 129.9 | 150.7 | 133.9 | 133.4 | 119.7 | 113.5 |
| UNg 7 | 389.4 | 449.9 | 447 | 405.9 | 404.1 | 474.9 | 314.3 |
| UNg 8 | 80.9 | 79.3 | 118.2 | 98.4 | 86.9 | 152 | 66.1 |

**Table 7.** Activity concentration of $^{232}$Th (Bq/kg) isotopes of albite granite samples from Um Naggat, Central Eastern Desert, Egypt.

| Samples | $^{228}$Ac | $^{208}$Tl | $^{212}$Bi |
|---|---|---|---|
| UNg 1 | 632.1 | 668.8 | 720.9 |
| UNg 2 | 251.5 | 269.2 | 288.6 |
| UNg 3 | 178 | 190.2 | 208.3 |
| UNg 4 | 188.9 | 194.4 | 213.7 |
| UNg 5 | 252 | 262.1 | 298 |
| UNg 6 | 114.5 | 118.9 | 131.8 |
| UNg 7 | 54.7 | 56.9 | 63.2 |
| UNg 8 | 84.6 | 94.5 | 100.7 |

In Table 8, the concentrations of the radioelements in all the samples from the Um Naggat area are higher than the worldwide average, except $^{235}$U, which is lower than the worldwide average, while samples UNg 1 and UNg 5 are higher than the worldwide average.

**Table 8.** Activity concentration of $^{238}$U, $^{235}$U, $^{226}$Ra, $^{232}$Th, and $^{40}$K (Bq/kg) of albite granite samples from Um Naggat, Central Eastern Desert, Egypt.

| Sample | $^{238}$U | $^{226}$Ra | $^{235}$U | $^{232}$Th | $^{40}$K | $^{238}$U/$^{235}$U | $^{226}$Ra/$^{238}$U | $^{232}$Th/$^{238}$U |
|---|---|---|---|---|---|---|---|---|
| UNg 1 | 785.1 | 844.5 | 36.3 | 673.9 | 567.1 | 21.6 | 1.1 | 0.86 |
| UNg 2 | 578.1 | 725.7 | 26.9 | 269.8 | 1218.7 | 21.4 | 1.3 | 0.47 |
| UNg 3 | 428.6 | 514.9 | 20 | 192.2 | 845.1 | 21.4 | 1.2 | 0.45 |
| UNg 4 | 359.9 | 345.3 | 16.6 | 199 | 1222.8 | 21.7 | 1 | 0.55 |
| UNg 5 | 800.4 | 1016.6 | 37.1 | 270.7 | 578.3 | 21.6 | 1.3 | 0.34 |
| UNg 6 | 127.7 | 150.7 | 6 | 121.7 | 1264.6 | 21.1 | 1.2 | 0.95 |
| UNg 7 | 419.7 | 447 | 19.3 | 58.3 | 1051.1 | 21.7 | 1.1 | 0.14 |
| UNg 8 | 80.1 | 118.2 | 3.8 | 93.3 | 2329.4 | 21.3 | 1.5 | 1.16 |
| Min | 80.1 | 118.2 | 3.8 | 58.3 | 567.1 | 21.1 | 1 | 0.14 |
| Max | 800.4 | 1016.6 | 37.1 | 673.9 | 2329.4 | 21.7 | 1.5 | 1.16 |
| Ave. | 447.5 | 520.4 | 20.8 | 234.9 | 1134.6 | 21.5 | 1.2 | 0.62 |
| W. A | 33 | 32 | 33 | 45 | 500 | 21.7 | 1 | 3.5 |

The activity ratio $^{226}$Ra/$^{238}$U was calculated for all the samples. It showed a disequilibrium between $^{226}$Ra and $^{238}$U, except in samples UNg 1, UNg 4, and UNg 7. The $^{238}$U/$^{235}$U activity ratio for all the samples varied between 21.1 and 21.7, with an average of 21.5, which is close to the worldwide average. The correlation curves of the activity concentrations between $^{226}$Ra and $^{232}$Th, $^{226}$Ra and $^{40}$K, and $^{232}$Th and $^{40}$K from the Um Naggat area are shown in Figure 6. The correlation between $^{226}$Ra and $^{232}$Th ($R^2 = 0.42$) is considered a positive relationship, as shown in Figure 6a, whereas the relationship between $^{40}$K and both $^{226}$Ra and $^{232}$Th is a negative correlation, with values of $R^2 = 0.60$ and $R^2 = 0.31$, respectively, as shown in Figure 6b,c. This also demonstrates the geological characteristics of the

granite rocks in the investigated location, where an alteration resulted in the appearance of heavy minerals such as uranothorite, uranophane autunite, and thorite. Rutile, samarskite, columbite, xenotime, monazitezircon, fluorite, and fergusonite are examples of uncommon metals. Meanwhile, the $^{232}$Th/$^{238}$U ratio for most samples was less than Clark's value (3.5), indicating and confirming the U-enrichment of the albite granite in the Um Naggat area.

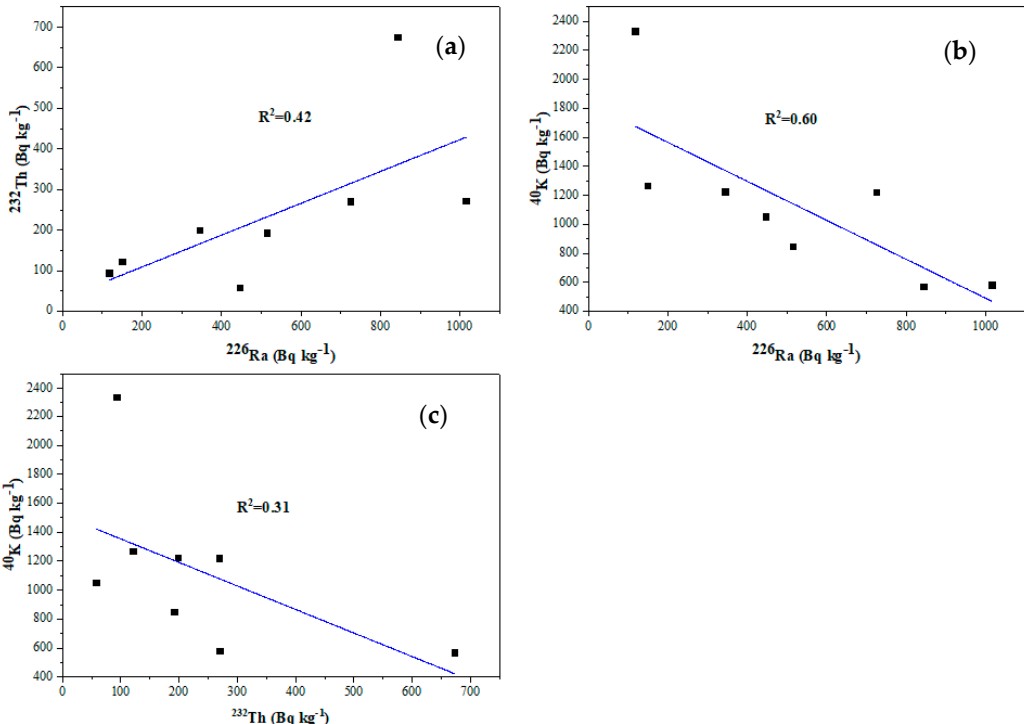

**Figure 6.** Correlation of the activity concentrations between: (**a**) $^{226}$Ra and $^{232}$Th, (**b**) $^{226}$Ra and $^{40}$K and (**c**) $^{232}$Th and $^{40}$K of albite granites at Um Naggat, Central Eastern Desert, Egypt.

*3.3. Radiation Health Hazards*

The assessment of the radiation doses received by humans from natural sources is of special importance because the absorbed dose by γ-ray exposure depends not only on the γ-ray energy but also on the material, owing to changes in its physical properties [32,33]. There is concern that some rocks transmit excessive radiation doses to the body due to the γ-rays emitted by the $^{232}$Th decay chain; the $^{214}$Pb and $^{214}$Bi progeny of $^{226}$Ra and $^{40}$K also contribute to the total body radiation dose. The absorbed dose rates one meter from the studied rocks due to their terrestrial radioactivity are calculated using the following equation;

$$D_{air} \text{ (nGy/h)} = K_{Ra} A_{Ra} + K_{Th} A_{Th} + K_K A_K \tag{2}$$

where $D_{air}$ is the absorbed dose rate (nGy/h) 1 m from the surface of the albite granite and $A_{Ra}$, $A_{Th}$, and $A_K$ represent the activity concentrations of Ra, Th, and K, respectively. Furthermore, $K_K$, $K_{Th}$, and $K_{Ra}$ are the conversion factors (or dose rate coefficients) expressed in (nGy/h per Bq/kg) for potassium (0.043), thorium (0.662), and radium (0.427), respectively [34].

However, direct measurements of absorbed dose rates in the air have been carried out in many countries around the world. The values of the absorbed dose rates 1 m from the surface of the albite granitic rocks at nine and eight locations distributed over Abu Rusheid and Um Naggat areas, respectively, are presented in Tables 9 and 10. The population weighted average is 59 nGy/h [3].

**Table 9.** The statistical analysis of the activity concentration of $^{226}$Ra, $^{232}$Th, and $^{40}$K, radium equivalent Ra$_{eq}$ (Bq/kg), external hazard index (H$_{ex}$), internal hazard index (H$_{in}$), representative gamma index (Iγ), absorbed dose rate, D$_{air}$ (nGy h$^{-1}$), outdoor annual effective dose, AED$_{out}$ (mSv), indoor effective dose, AED$_{in}$ (mSv), annual gonadal dose equivalent, AGDE (mSv), and excess lifetime cancer (ELCR) at Abu Rusheid, Central Eastern Desert, Egypt.

| Parameters | N | Average | SD | Min | Max |
|---|---|---|---|---|---|
| Ra$_{eq}$ | 9 | 1672 | 797 | 536 | 2760 |
| H$_{in}$ | 9 | 6 | 3 | 2 | 11 |
| H$_{ex}$ | 9 | 5 | 2 | 1 | 7 |
| I$_\gamma$ | 9 | 6 | 3 | 2 | 10 |
| D$_{air}$ | 9 | 740 | 344 | 250 | 1200 |
| AED$_{out}$ | 9 | 0.9 | 0 | 0 | 1 |
| AED$_{in}$ | 9 | 3.6 | 2 | 1 | 6 |
| AGDE | 9 | 5 | 2 | 2 | 8 |
| ELCR $\times 10^{-3}$ | 9 | 3 | 1 | 1 | 5 |

**Table 10.** The statistical analysis of the activity concentration of $^{226}$Ra, $^{232}$Th, and $^{40}$K, radium equivalent Ra$_{eq}$ (Bq/kg), external hazard index (H$_{ex}$), internal hazard index (H$_{in}$), representative gamma index (Iγ), absorbed dose rate, D$_{air}$ (nGy h$^{-1}$), outdoor annual effective dose, AED$_{out}$ (mSv), indoor effective dose, AED$_{in}$ (mSv), annual gonadal dose equivalent, AGDE (mSv), and excess lifetime cancer (ELCR) at Um Naggat, Central Eastern Desert, Egypt.

| Parameters | N | Mean | SD | Min | Max |
|---|---|---|---|---|---|
| Ra$_{eq}$ | 8 | 944 | 514 | 422 | 1852 |
| H$_{in}$ | 8 | 4 | 2 | 1 | 7 |
| H$_{ex}$ | 8 | 3 | 1 | 1 | 5 |
| I$_\gamma$ | 8 | 3 | 2 | 2 | 6 |
| D$_{air}$ | 8 | 429 | 225 | 195 | 820 |
| AED$_{out}$ | 8 | 0.5 | 0 | 0 | 1 |
| AED$_{in}$ | 8 | 2.1 | 1 | 1 | 4 |
| AGDE | 8 | 3 | 2 | 1 | 6 |
| ELCR $\times 10^{-3}$ | 8 | 2 | 1 | 1 | 4 |

The absorbed dose rate D$_{air}$ (nGy/h) for the Abu Rusheid samples was higher than the worldwide average 59 (nGy/h), as shown in Table 9. The minimum and maximum values were 200 and 1200 nGy/h, respectively, with an average of 740 nGy/h. This reveals that the average absorbed dose rate in the air 1 m from the surface of the studied albite granite was much higher than the worldwide average. The absorbed dose rate D$_{air}$ for all the Um Naggat samples (Table 10) was lower than that of the Abu Rusheid samples, but the average value of D$_{air}$ (429 nGy/h) was lower than the worldwide average. The D$_{air}$ values varied from 195 to 820 nGy/h.

The annual effective dose (AED) was calculated from the absorbed dose (D) by applying the dose conversion factor of 0.7 Sv/Gy and the occupancy factor (OC) of 0.2 and 0.8 for outdoor and indoor exposures, respectively, through the exposure time 8760 h per year [35].

$$\text{AED (mSv y}^{-1}) = D(\text{nGy/h}) \times 8760(\text{h}) \times \text{OC} \times 0.7 \text{ (Sv/Gy)} \times 10^{-6} \tag{3}$$

Tables 9 and 10 illustrate that all the total average values of AED obtained from the examined samples were higher than the worldwide annual effective dose 1 mSvy$^{-1}$ [35]. The mean values of AED$_{out}$ were 0.9 and 0.5 mSv y$^{-1}$ for Abu Rusheid and Um Naggat albite granite, respectively, while the AEDs were 3.6 and 2.1 for Abu Rusheid and Um Naggat, respectively. Thus, indoor exposure can cause dangerous health effects.

The activity levels of $^{226}$Ra, $^{232}$Th, and $^{40}$K in the examined granitic samples can be determined using the radium equivalent activity (Ra$_{eq}$) index. This index can be calculated using the following equation [36]:

$$Ra_{eq} \ (Bq/kg) = A_{Ra} + 1.43A_{Th} + 0.077A_K \tag{4}$$

The average obtained values were 1672 and 944 Bq·kg$^{-1}$ from the samples from Abu Rusheid and Um Naggat, respectively. The average was five and three times higher than the recommended limit (370 Bq·kg$^{-1}$) (see Tables 9 and 10), which keeps the external dose below 1.5 mSv·y$^{-1}$ [37]. This illustrates that the studied samples are not safe to apply in the building materials and the infrastructure fields.

External and internal hazard indices (H$_{ex}$ and H$_{in}$) were utilized to compute the rate of radiation dose released by natural terrestrial radionuclides in the investigated samples [38]. Furthermore, a group of experts suggested another index that could be used to estimate the level of γ-radiation hazard associated with the natural radionuclides in the samples due to the different combinations of specific natural activities in the sample [38].

$$H_{ex} = A_{Ra}/370 + A_{Th}/259 + A_K/4810 \tag{5}$$

$$H_{in} = A_{Ra}/185 + A_{Th}/259 + A_K/4810 \tag{6}$$

$$I_\gamma = \frac{A_{Ra}}{150} + \frac{A_{Th}}{100} + + \frac{A_K}{1500} \tag{7}$$

Tables 9 and 10 demonstrate that the average values of H$_{ex}$ and H$_{in}$ in all the measured stations of albite granite samples at Abu Rusheid (six and five, respectively) and Um Naggat (four and three, respectively) are observed to be higher than unity (Figure 7). The H$_{ex}$ and H$_{in}$ values reveal that the albite granites would exert adverse health effects linked to external gamma radiation and radon gas and its decay products [6]. The data average varied from 3 to 6 at Um Naggat and Abu Rusheid, respectively. The I$_\gamma$ index is higher than the permissible limit compared with unity (Figure 7).

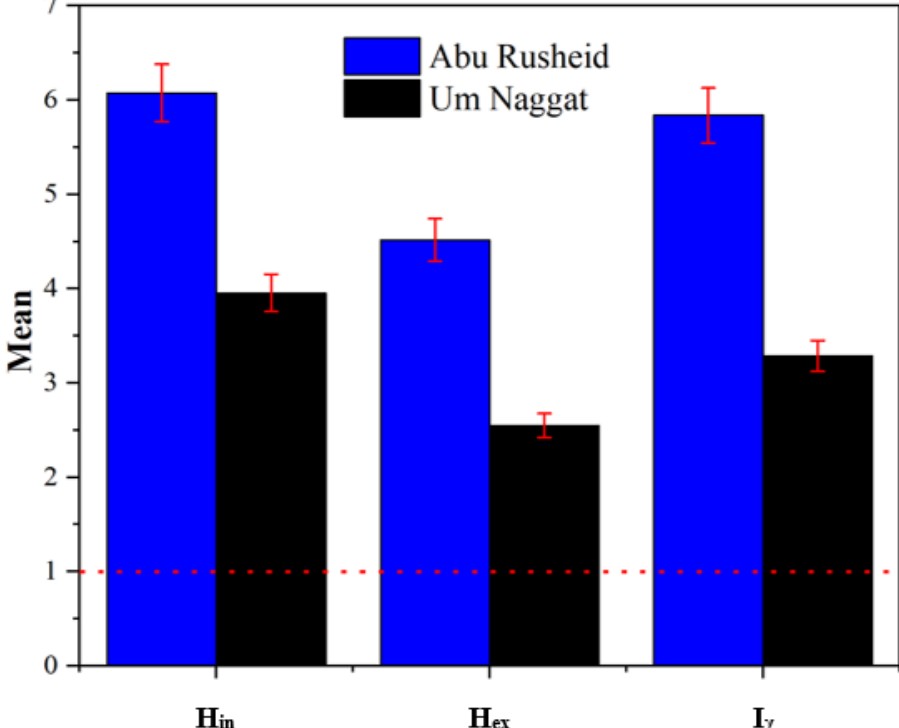

**Figure 7.** Mean values of the external (H$_{ex}$) and internal (H$_{in}$) hazard indices and representative gamma index (I$_\gamma$) of the albite granites in the Abu Rusheid and Um Naggat areas.

The annual gonadal dose equivalent (AGDE) is the radiological factor used to detect the associated dose of organs per year for people, especially the gonads. The date of AGDE is computed based on the gamma radiation released from the natural radionuclides and is computed from the Equation (8) [39]:

$$\text{AGDE (mSv·y}^{-1}) = 3.09 A_{Ra} + 4.18 A_{Th} + 0.314 A_{K} \tag{8}$$

Tables 9 and 10 reveal the statistical results of AGDE calculated for all the albite granites in the examined areas. The AGDE values exceeded the permissible limit of 0.3 mSv·y$^{-1}$ [40]. At Abu Rusheid, the range of AGDE values was 2 mSv·y$^{-1}$ to 8 mSv·y$^{-1}$ with an average value of 5 mSv·y$^{-1}$ (Table 9), while the minimum and maximum values were 1 mSv·y$^{-1}$ and 6 mSv· y$^{-1}$, with the average an value of 3 mSv·y$^{-1}$ in the Um Naggat area (Table 10). Thus, the application of albite granites in building materials would entail significant health risks.

The excess lifetime cancer risk (ELCR) is the radioactive parameter is applied to predict the cancer risk associated with long exposure to albite granites. The following equation can be utilized to compute ELCR [41]:

$$\text{ELCR} \left( \text{mSv·y}^{-1} \right) = \text{AED}_{out} \times \text{DL} \times \text{RF} \tag{9}$$

The computation depends on the outdoor annual effective dose (AED$_{out}$), the duration of life (DL, 70 years), and the risk factor for cancer (RF, 0.05 Sv$^{-1}$), as recommended by ICRP (International Commission of Radiation Protection).

Tables 9 and 10 display the ELCR date for the albite granite samples detected at Abu Rusheid and Um Naggat. The ELCR values ranged from $1 \times 10^{-3}$ to $5 \times 10^{-3}$, with a mean value of $3 \times 10^{-3}$ (Table 9), in Abu Rusheid, while the values ranged from $1 \times 10^{-3}$ to $4 \times 10^{-3}$, with an average value $2 \times 10^{-3}$ in Um Naggat (Table 10). This value is higher than the limit of 0.00029 of the worldwide average [42]. The ELCR values are more significant than the international mean value and suggest that long exposure to these albite granites may cause cancers and other serious diseases.

## 4. Statistical Analysis

*Cluster Analysis*

Because of the large number of parameters in the correlation analysis data, the analysis appears to be complicated. However, the correlations between the radioactive characteristics can be recognized and exposed qualitatively using hierarchical cluster analysis (HCA). HCA is a data classification system that uses multivariate algorithms to determine real data groups. Objects are grouped in such a way that they all belong to the same category. The results with the highest degree of nearness are categorized first in hierarchical clustering, followed by the next most similar data. The process is continued until all of the information has been classified. The degrees of similarity at which the data mix are used to create a dendrogram. A similarity of 100% indicates that the clusters are divided from comparable sample measures by zero distance, whereas a similarity of 0% indicates that the clustering regions are as unlike as the least similar region. In this work, Ward's strategy of cluster analysis was applied. Ward's method is a connection procedure for estimating the Euclidean distance between the activity concentrations of radionuclides and radiological parameters [43] (Figure 8a,b). Three clusters were plotted in the dendrogram of the examined results of the two studied areas. At Abu Rusheid, cluster I consisted of $^{226}$Ra, while cluster II included $^{232}$Th and the corresponding radiological hazard parameters. At the same time, Cluster III included the $^{40}$K activity concentration. Moreover, at Um Naggat, cluster I contained $^{226}$Ra and the corresponding radiological parameters, while cluster II included the $^{232}$Th and $^{40}$K activity concentration observed in cluster III. Thus, it can be concluded that the radioactivity and radiation exposure of albite granites was linked mainly to the radium and thorium activity concentrations.

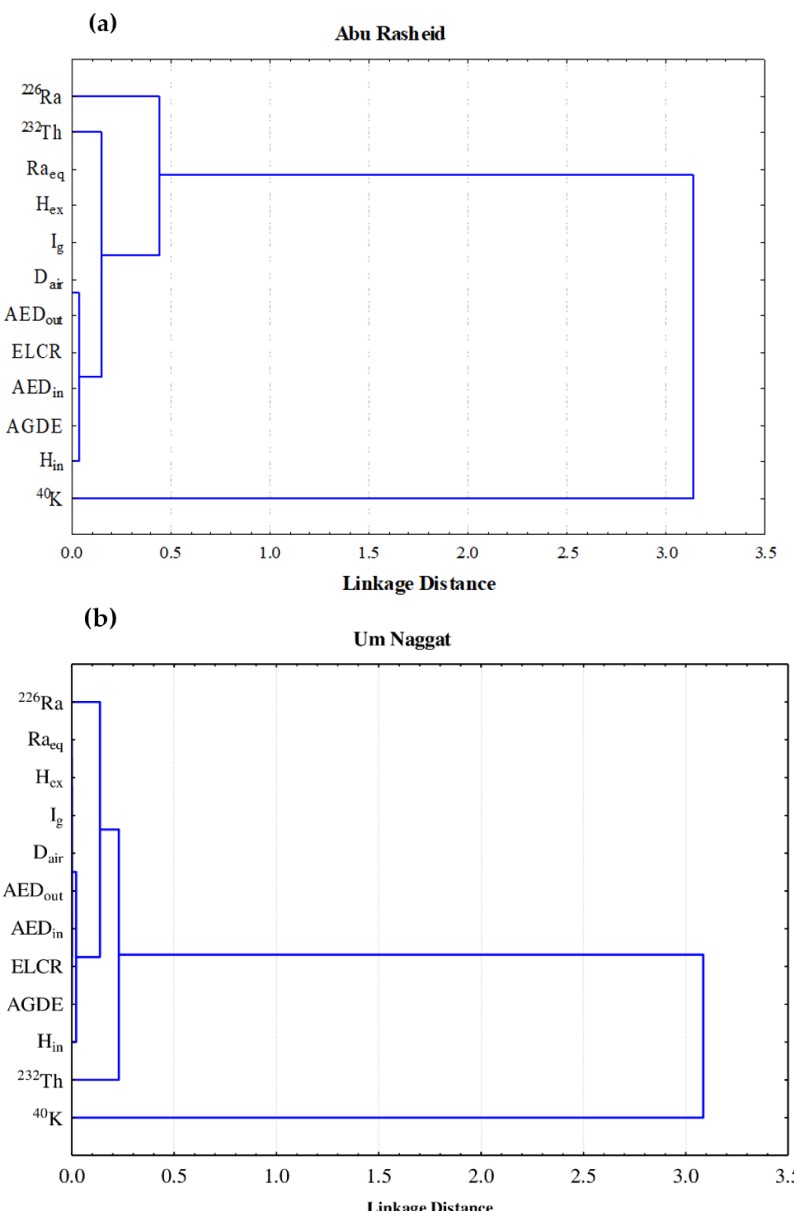

**Figure 8.** The clustering analysis of the radiological parameters of the albite granites in (**a**) Abu Rusheid and (**b**) Um Naggat.

## 5. Conclusions

This study aimed to assess the radioactivity released from albite granite rocks, which may be used in different infrastructure applications. A statistical analysis was performed to show the geological processes that are thought to increase the radioactive contents in the albite granite. The activity concentrations estimated for $^{238}$U (range from 204 to 1127 Bq/kg), $^{226}$Ra (range from 215 to 1300 Bq/kg), $^{232}$Th (from 130 to 1424 Bq/kg), and $^{40}$K (from 1108 to 2167 Bq/kg) in the Abu Rusheid area, as well as 238U (range from 80 to 800 Bq/kg), $^{226}$Ra (range from 118 to 1017 Bq/kg), $^{232}$Th (from 58 to 674 Bq/kg), and $^{40}$K (from 567 to 2329 Bq/kg) in the Um Naggat area, were significantly higher than the worldwide average values. Furthermore, the radiological hazard factors were estimated in the albite granite samples and found to be higher than the approved levels. This is linked to the alteration in the radioactive minerals and rare metals in the studied albite granites. Therefore, the albite granites in the investigated areas exert adverse health effects and cannot be utilized in building materials and numerous infrastructure fields.

**Author Contributions:** I.G. and M.Y.H. contributed equally to this work. Conceptualization, I.G., M.E. and M.Y.H.; methodology, I.G. and M.E.; software, M.E. and M.Y.H.; validation, M.U.K. and D.A.B.; formal analysis, I.G., M.E. and M.Y.H.; investigation, M.U.K. and M.I.S.; resources, I.G. and M.E.; data curation, M.E. and M.Y.H.; writing—original draft preparation, I.G., M.E. and M.Y.H.; writing—review and editing, I.G., M.U.K. and M.Y.H.; visualization, M.I.S. and D.A.B.; supervision, I.G., M.U.K. and M.I.S.; project administration, I.G. and M.E.; funding acquisition, A.S., N.T. All authors have read and agreed to the published version of the manuscript.

**Funding:** The authors express their gratitude to Princess Nourah bint Abdulrahman University Researchers Supporting Project (Grant No. PNURSP2022R12), Princess Nourah bint Abdulrahman University, Riyadh, Saudi Arabia.

**Institutional Review Board Statement:** Not applicable.

**Informed Consent Statement:** Not applicable.

**Data Availability Statement:** Not applicable.

**Acknowledgments:** The authors express their graduate to the Nuclear Materials Authority. The authors express their gratitude to Princess Nourah bint Abdulrahman University Researchers Supporting Project (Grant No. PNURSP2022R12), Princess Nourah bint Abdulrahman University, Riyadh, Saudi Arabia.

**Conflicts of Interest:** The authors declare no conflict of interest.

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
