# Peer review of "Assessment of Radioactive Materials in Albite Granites from Abu Rusheid and Um Naggat, Central Eastern Desert, Egypt"

_minerals, doi:10.3390/min12020120_

Round 1

Reviewer 1 Report

The paper reports on the content of 238U, 226Ra, 232Th, and 40K in two areas of the Central Eastern Desert, Egypt and the intensity of gamma-radiation generated by the radioactive decay of these elements. Not being an expert in this field, I cannot appreciate the practical importance of the presented results. However, the presentation of these results in the article requires improvement.

  1. The measurements were fulfilled for 17 locations, not samples.
  2. How these locations were selected?
  3. All captions to figures and tables should clearly explain what is shown on the axes or listed in the columns.
  4. The same remark applies to the terms of equations 2-8.
  5. What can be learned from the correlations shown in Figures 5 and 6? Why are these correlations not discussed?
  6. Explain in detail what is shown in Figure 8.

Author Response

09 January 2022

Dear Reviewer,

Please find attached the submission of the carefully revised version of the manuscript in Ref., following the minor comments and modification of the Reviewer.

Below is a detailed list of the changes made in response to the Reviewer’s minor comments (in italics), which outlines every change made a point by point. The changes are marked in the manuscript text (yellow highlighted).

  • The measurements were fulfilled for 17 locations, not samples.

The measurements are performed in the field using the calibrated detector.

  • How were these locations selected?

The locations are selected based on the importance of surrounding areas for the studied area where the surrounding areas are used to living activities of people.

  • All captions to figures and tables should clearly explain what is shown on the axes or listed in the columns.

The captions are corrected to figures and tables.

  • The same remark applies to the terms of equations 2-8.

All uncleared terms of the equations are described in the manuscript.

  • What can be learned from the correlations shown in Figures 5 and 6? Why are these correlations not discussed?

The correlations shown in Figures 5 and 6 are described in detail in the manuscript.

  • Explain in detail what is shown in Figure 8.

The cluster analysis is explained in the manuscript.

We thank the Reviewer a lot for the useful and valuable comments that have helped improve the manuscript.

Hoping that all the careful review is sufficient for the direct acceptance of the manuscript, thank you for your time and consideration.

Best wishes,

Mohamed. Y. M. Hanfi

on behalf of all co-authors

Reviewer 2 Report

This work presents an assessment of radioactive materials in albite granites from Abu  Rusheid and Um Naggat, Central Eastern Desert, Egypt Rusheid and Um Naggat, Central Eastern Desert, Egypt. The work is comprehensive and important.  Following suggestions are provided to improve it.

1) The Introduction parts needs to be improved by describing the current research status of the relevant work. 

2) The methodology and equipment used in this work should be provided in Section 2. However, it looks like the authors described the methodologies in Section 3.

3) In the caption of Figure 3 and 4, (a),(b) and (c) should be described separately.   

4) In Figure 3,  what are the X, Y axis representing for? Same issue for Figure 4.

5) In Figure 5 and Figure 6, why did the author try to use a linear function to describe the relationship?   

6) Some of the deviations in Figure 5 and Figure 6 are large, please explain the large  data vibration. How many tests for each data point? Are they averaged?

7) Please check the format of Figure 8. Add the information for X, Y axis in the figure.

Author Response

09 January 2022

Dear Reviewer,

Please find attached the submission of the carefully revised version of the manuscript in Ref., following the minor comments and modification of the Reviewer.

Below is a detailed list of the changes made in response to the Reviewer’s minor comments (in italics), which outlines every change made a point by point. The changes are marked in the manuscript text (green highlighted).

Comments:

1) The Introduction parts needs to be improved by describing the current research status of the relevant work.

The introduction improved in the manuscript.

2) The methodology and equipment used in this work should be provided in Section 2. However, it looks like the authors described the methodologies in Section 3.

The methodology and equipment are modified in section 2.

3) In the caption of Figure 3 and 4, (a), (b) and (c) should be described separately.  

a, b and c are described separately in the captions of figures.

4) In Figure 3, what are the X, Y axis representing for? Same issue for Figure 4.

The figures are corrected in the manuscript. X, Y are identified.

5) In Figure 5 and Figure 6, why did the author try to use a linear function to describe the relationship?  

The linear function helps us to find the answer to the following question: “what are the radioactive sources in the albite granites?”

6) Some of the deviations in Figure 5 and Figure 6 are large, please explain the large data vibration. How many tests for each data point? Are they averaged?

The deviations show the correlations between activity concentrations of 226Ra, 232Th and 40K are strong, moderate and weak. The description of correlation added in detail in the manuscript. The radioactivity in each sample measured using the calibrated gamma spectrometry HPGe.   

7) Please check the format of Figure 8. Add the information for X, Y axis in the figure.

Format Figure 8 corrected in the manuscript. Already, the information of X and Y in the figure was added.

We thank the Reviewer a lot for the useful and valuable comments that have helped improve the manuscript.

Hoping that all the careful review is sufficient for the direct acceptance of the manuscript, thank you for your time and consideration.

Best wishes,

Mohamed. Y. M. Hanfi

on behalf of all co-authors

Reviewer 3 Report

I appreciate the authors for the study which can help the community. They have presented the paper in a readable manner. The following can be included so that the reader can have a good guidance from the paper.

  1. I suggest that the authors can include the instrumentation part about HPGe spectrometer and Gamma-ray spectrometry
  2. Check the equation, units and representation (for Example in page 13 author have used mSvy-1 while in the majority of occasion they have used mSv/y). Uniformity can be maintained in all the notations.
  3. I suggest adding the error bars to Figure 7 for more clarity on the measurement carried out.
  4. In the analysis of the statistical data, 4.1, the scientific interpretation(reason) can be provided by the authors
  5. Conclusion Part – The authors say that these materials are not utilized for the construction but they started the article saying that these materials are used for construction so that the interior radiation is more than that of outside radiation. Hence, I suggest the authors to make their point clear that are they suggesting that these materials should not be used or they say that these materials are already not in use? Kindly confirm.

Author Response

09 January 2022

Dear Reviewer,

Please find attached the submission of the carefully revised version of the manuscript in Ref., following the minor comments and modification of the Reviewer.

Below is a detailed list of the changes made in response to the Reviewer’s minor comments (in italics), which outlines every change made a point by point. The changes are marked in the manuscript text (Turquoise highlighted).

Comments:

I suggest that the authors can include the instrumentation part about HPGe spectrometer and Gamma-ray spectrometry

The instrumentation part about HPGe was added in the manuscript.

Check the equation, units and representation (for Example in page 13 author have used mSvy-1 while in the majority of occasion they have used mSv/y). Uniformity can be maintained in all the notations.

Uniformity was maintained in all the notations.

I suggest adding the error bars to Figure 7 for more clarity on the measurement carried out.

The error bars are added in Figure 7.

In the analysis of the statistical data, 4.1, the scientific interpretation(reason) can be provided by the authors

The scientific interpretation is provided in the manuscript.

Conclusion Part – The authors say that these materials are not utilized for the construction but they started the article saying that these materials are used for construction so that the interior radiation is more than that of outside radiation. Hence, I suggest the authors to make their point clear that are they suggesting that these materials should not be used or they say that these materials are already not in use? Kindly confirm.

Thanks for your recommendation. The correction of the conclusion part was edited in the manuscript.

We thank the Reviewer a lot for the useful and valuable comments that have helped improve the manuscript.

Hoping that all the careful review is sufficient for the direct acceptance of the manuscript, thank you for your time and consideration.

Best wishes,

Mohamed. Y. M. Hanfi

on behalf of all co-authors

Round 2

Reviewer 1 Report

The reviewer's comments have been answered satisfactorily.

Author Response

Comment: The reviewer's comments have been answered satisfactorily.

Response: Thank you very much. Your comments are helped us to improve the manuscript.

Reviewer 2 Report

The reviewer is satisfied with the replies except for the introduction part. It would be better if the introduction part is further enriched to strengthen the current status of this work.

Author Response

15 January 2022

Dear Reviewer,

Please find attached the submission of the carefully revised version of the manuscript in Ref., following the minor comments and modification of the Reviewer.

Below is a detailed list of the changes made in response to the Reviewer’s minor comments (in italics), which outlines every change made a point by point. The changes are marked in the manuscript text (green highlighted).

Comment: The reviewer is satisfied with the replies except for the introduction part. It would be better if the introduction part is further enriched to strengthen the current status of this work.

Response: The modifications and the corrections are added to the manuscript.

We thank the Reviewer a lot for the useful and valuable comments that have helped improve the manuscript.

Hoping that all the careful review is sufficient for the direct acceptance of the manuscript, thank you for your time and consideration.

Best wishes,

Mohamed. Y. M. Hanfi

on behalf of all co-authors

This manuscript is a resubmission of an earlier submission. The following is a list of the peer review reports and author responses from that submission.